# Identification of HDV-like *theta* ribozymes involved in tRNA-based recoding of gut bacteriophages

Kasimir Kienbeck [1,3], Lukas Malfertheiner [2,3], Susann Zelger-Paulus [1], Silke Johannsen [1], Christian von Mering[2] ✉ & Roland K. O. Sigel [1] ✉

Trillions of microorganisms, collectively known as the microbiome, inhabit our bodies with the gut microbiome being of particular interest in biomedical research. Bacteriophages, the dominant virome constituents, can utilize suppressor tRNAs to switch to alternative genetic codes (e.g., the UAG stop-codon is reassigned to glutamine) while infecting hosts with the standard bacterial code. However, what triggers this switch and how the bacteriophage manipulates its host is poorly understood. Here, we report the discovery of a subgroup of minimal hepatitis delta virus (HDV)-like ribozymes – *theta* ribozymes – potentially involved in the code switch leading to the expression of recoded lysis and structural phage genes. We demonstrate their HDV-like self-scission behavior in vitro and find them in an unreported context often located with their cleavage site adjacent to tRNAs, indicating a role in viral tRNA maturation and/or regulation. Every fifth associated tRNA is a suppressor tRNA, further strengthening our hypothesis. The vast abundance of tRNA-associated *theta* ribozymes – we provide 1753 unique examples – highlights the importance of small ribozymes as an alternative to large enzymes that usually process tRNA 3'-ends. Our discovery expands the short list of biological functions of small HDV-like ribozymes and introduces a previously unknown player likely involved in the code switch of certain recoded gut bacteriophages.

Ribozymes are ubiquitous and participate in essential biological processes in all domains of life, including peptidyl transferase activity[1] and the transesterification steps required for tRNA maturation[2] as well as eukaryotic mRNA splicing[3]. Small ribozymes (<200 nucleotides; nt) are restricted to self-cleavage and/or -ligation but are remarkably diverse in sequence, structure, and biological functions[4–8]. A well-studied example is the family of HDV-like ribozymes (delta-like ribozymes, drzs), which have a highly conserved, nested double-pseudoknotted structure but considerable variability in primary sequence[9–11] (Fig. 1a). While biological functions of specific drz examples are known[12–17], the majority, especially minimal variants, are less understood. Minimal

drzs, which were first identified in metagenomic samples, lack the P4 domain[18] (Fig. 1a) and their origin (eukaryotic, bacterial, or viral) and biological functions remain to be determined.

Herein, we report the discovery and in vitro validation of minimal drzs within *Caudoviricetes* bacteriophage genomes of the mammalian gut often associated with viral tRNAs and designate them as *theta* ribozymes (Θrzs). The gut virome is mainly composed of bacteriophages (>90%) and is increasingly linked to human health and disease[19–22], leading to multiple sequence database additions in recent years. Sequence analyses of these databases have revealed that some bacteriophages are recoded: they use genetic codes in which a certain

[1]Department of Chemistry, University of Zurich, Zurich CH-8057, Switzerland. [2]Department of Molecular Life Sciences and Swiss Institute of Bioinformatics, University of Zurich, Zurich CH-8057, Switzerland. [3]These authors contributed equally: Kasimir Kienbeck, Lukas Malfertheiner. ✉e-mail: christian.von.mering@mls.uzh.ch; roland.sigel@chem.uzh.ch

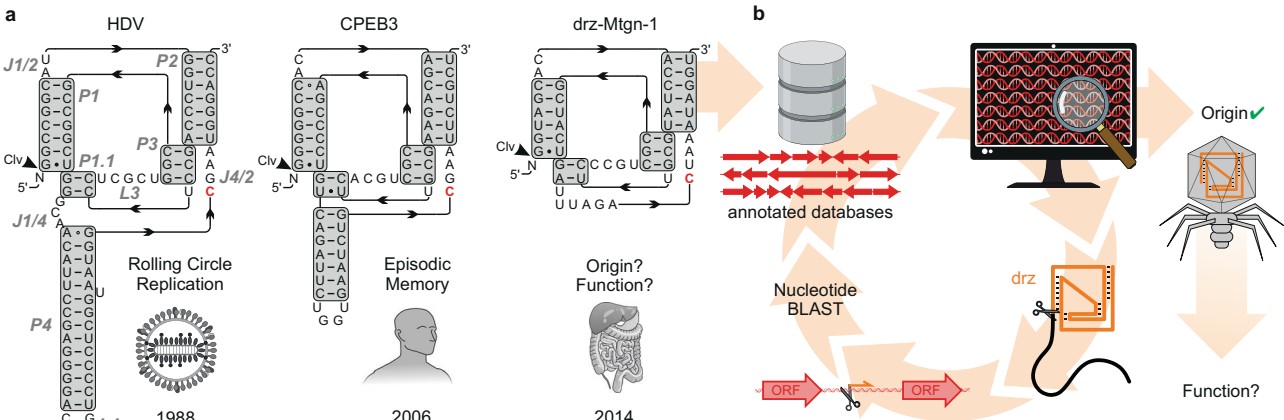

**Fig. 1 | HDV-like ribozymes (drzs) and their association with bacteriophages.**
**a** Secondary structure of representative drzs: HDV[12], the human CPEB3 ribozyme[62], and the metagenomic drz "drz-Mtgn-1" with unknown origin and function[18]. Helical domains are highlighted in gray, domain labels in gray italics, catalytic cytosine residues are marked in red. **b** Workflow of the initial nucleotide-based drz homology search: basic local alignment search tool (BLAST[26]) analysis recovered drz-Mtgn-1 from annotated bacteriophage databases, subsequent BLAST analyses of nearby open reading frames (ORF) revealed multiple drzs (Supplementary Fig. 1). Colors: viral DNA and proteins: red; drz: orange.

stop-codon is reassigned to a standard amino acid[23]. One example, where the amber stop-codon (UAG) is recoded to glutamine (code 15), was recently experimentally verified[24]. Viral suppressor tRNAs (tRNA[Sup]) are central players in the translation of alternative genetic codes, a tool which may be used by recoded phages to initiate host lysis. However, the precise mechanism of the lytic-lysogenic switch in bacteriophages is not yet fully characterized.

In this study, we propose that tRNA-associated Ɵrzs are involved in viral tRNA maturation and may support expression of late-phase lysis and structural genes containing recoded stop-codons in a subset of recoded bacteriophages, potentially even triggering phage lysis. Our findings provide insights into the intriguing world of drzs and their biological significance.

## Results

### The viral origin of metagenomic minimal drzs

The minimal drz "drz-Mtgn-1" was identified in a metagenomic sample[18], but its origin and biological function remain unclear (Fig. 1a). We were intrigued by this knowledge gap and the drz's unique behavior with divalent metal ions, and conducted a nucleotide sequence-based search of publicly available sequence databases[25]. This search ultimately led, among others, to the assignment of drz-Mtgn-1 to several double-stranded DNA (dsDNA) bacteriophages (*Caudoviricetes*, Fig. 1b).

Due to the conservation of drz secondary structure rather than primary sequence, we changed our approach from using a basic local alignment search tool (BLAST[26]) with a published drz sequence to a motif-based search with RNArobo[27] using the minimal motif by ref. 18. Initial searches resulted in over 60 minimal drz sequences in bacteriophage genome databases[28,29] and three conclusions: (i) minimal drz sequences were initially detected exclusively in bacteriophage genomes assembled from human gut metagenomic data, (ii) the hits showed associations with nearby open reading frames (ORF; mostly of putative proteins with unknown functions), potentially providing insights into their biological functions (Supplementary Fig. 1), and (iii) minimal drzs are more widespread than previously thought, with dozens of hits discovered in an initial search of two databases compared to a few hits from a full-scale search conducted in 2014[18].

### Discovery of tRNA-associated *theta* ribozymes (Ɵrzs)

A subset of hits within our initial categorization captured our attention, specifically, minimal drzs adjacent to phage tRNA genes. The position of the ribozyme cleavage site (G1; Fig. 1a) at the 3′-end of the tRNA suggests a previously unknown biological function in tRNA maturation. We therefore focused on these examples and refined our search motif accordingly. We chose eight recently annotated viral databases[23,29–35] from diverse environments (Supplementary Table 1) and cross-referenced all subsequent hits with tRNA motif searches in the same databases. To increase motif specificity, we incorporated false-positive motifs as internal controls, considering previous findings that drzs are inactivated by a cytosine-to-uracil mutation (CΔU) in the active site[36], with no observed rescue mutation at this position[37]. Each search was conducted with four different descriptor files, one active ribozyme motif with the catalytic cytosine residue intact (first position in the J4/2 junction) and three false-positive motifs containing substitutions at this residue (CΔA, CΔG, and CΔU, respectively; Fig. 2a). An initial search yielded less than 50% of the total hits with the active motif, indicating a high false positive rate (Fig. 2b (i)). We iteratively improved the motif by shortening the L4 loop and J1/2 junction (Fig. 2b (i), (ii)) and applying nucleotide identity constraints based on preliminary consensus sequences (>97% conservation; Fig. 2b (iii)). Finally, we introduced one additional degree of freedom at the last position of the J4/2 junction in line with the HDV-like structural motif[10]. The latter reduced the false positive rate to nearly zero while increasing the number of detected tRNA-associated ribozymes (Fig. 2b (iv)). Sequences obtained from this refined motif are referred to as *theta* ribozymes (Ɵrz) due to their frequent associations with tRNAs.

Using the optimized motif, we identified 302 unique Ɵrz sequences in the above-mentioned databases, with 126 classified as tRNA-associated Ɵrzs, where the ribozyme's cleavage site is within ±5 nt of the 3′-end of a tRNA. Our analysis revealed 152 distinct Ɵrz-adjacent tRNA sequences, resulting in 185 unique tRNA/Ɵrz combinations. Many Ɵrzs are present in multiple viral genomes, totaling 1281 occurrences, with 742 (58%) classified as tRNA-associated Ɵrzs (Fig. 2c, top). 568 of these tRNA-associated Ɵrzs (77%) are directly adjacent to the respective tRNA (±1 nt). For the remainder (23%), we assume that slight inaccuracies in the computational prediction of tRNA ends explain the few additional nucleotides between the tRNA and the Ɵrz cleavage site.

Over 80% of Ɵrzs were found in human or animal gut bacteriophages, with no or only a few hits in phages isolated from other environments (Supplementary Table 1). To validate that the enrichment in gut-associated phage genomes is unbiased, we included

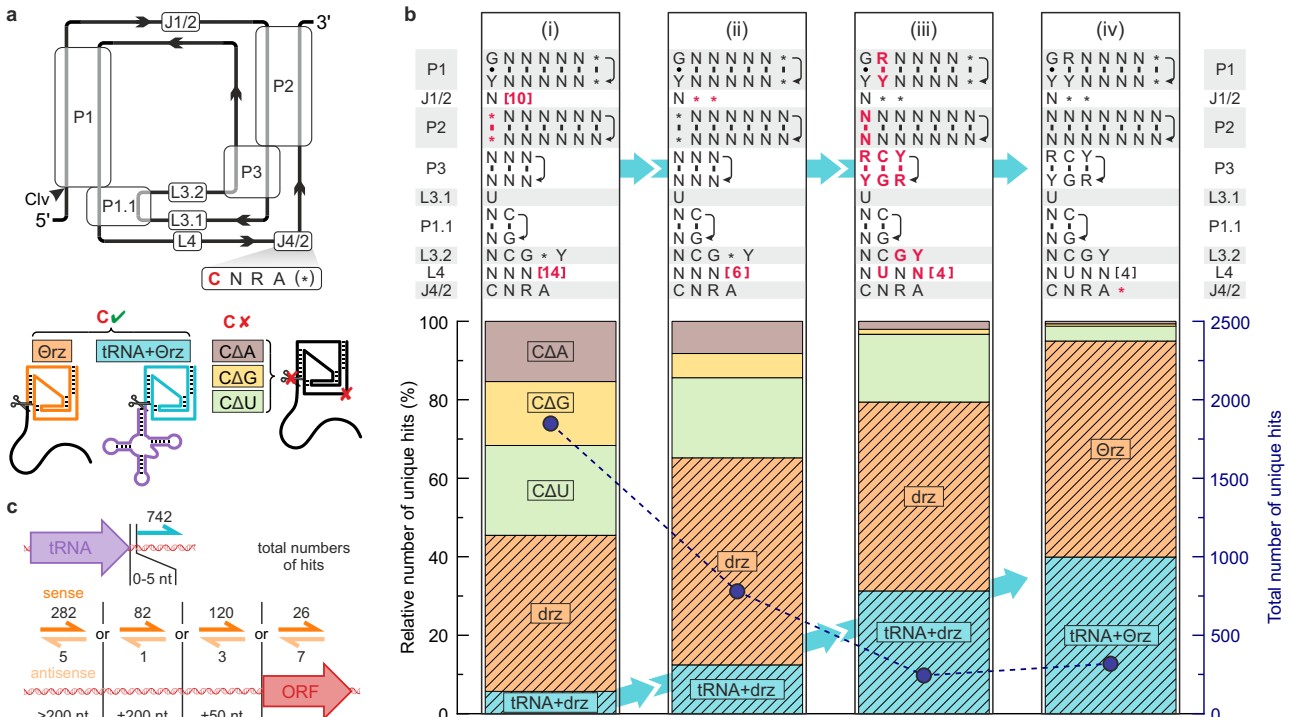

**Fig. 2 | Optimization of the *theta* ribozyme (Θrz) motif using annotated bacteriophage databases. a** Secondary structure of minimal drzs. The catalytic cytosine (red) in the J4/2 junction was substituted with A (CΔA: brown), G (CΔG: yellow), and U (CΔU: green), resulting in inactive motifs (false positives). True positives were subdivided into isolated ribozymes (orange) and tRNA-associated ribozymes (turquoise). **b** Changes to the search motif (top) in four steps (i)–(iv) and hits from annotated bacteriophage databases (bottom). Top: Domains denominated as in (**a**), helical strand directionality is indicated with black arrows. Modifications to the previous motif are indicated in bold red. (i) shows changes to the motif by ref. 18. IUPAC nucleotide denominations. * = any or no nucleotide. [X] = between zero and X nucleotides. Bottom: Relative number of unique hits with the motif above (left *y* axis). Putatively active ribozyme hits are striped. Total hit numbers shown in marine blue (right *y* axis; points and dashed line). Ribozyme sequences obtained by the final motif (iv) are denominated as Θrzs. **c** Top: Total tRNA-associated Θrzs (turquoise half-arrows) adjacent to viral tRNAs (purple). Bottom: Total number of isolated Θrz hits (orange half-arrows and black numbers) in a sense (dark orange) or antisense (light orange) to the closest ORF (red arrow). Source data are provided as a Source Data file.

bacterial[38] and eukaryotic genomes[39] (human, mouse, and protists) in our analysis. Although these databases are more than twice the size, the search revealed only 12 Θrz hits in bacterial genomes, none of which were associated with a tRNA. Consistent with our initial manual searches, nearly 99% of Θrz and minimal drz sequences were confined to bacteriophages belonging to the *Caudoviricetes* class, the predominant dsDNA viruses in the human gut virome[40] (Supplementary Fig. 2).

In silico host predictions were extracted from the databases, when available, resulting in two phyla: The predominant fraction (~90%) of bacterial hosts belongs to the *Bacteroidota* phylum, whereas the remainder is part of the *Bacillota*. Since some members of these phyla only sporadically encode a CCA-tail in their tRNAs[41], we analyzed the phage tRNAs for the presence of this essential feature. Interestingly, only 4.5% of all tRNAs associated with Θrzs encode for a CCA-tail. However, we found that 20.6% of Θrz-containing phage genomes encode for a tRNA adenylyltransferase, enabling post-transcriptional addition of a CCA-tail to tRNAs. In contrast, only 0.08% of all analyzed phages carry this enzyme, i.e., this gene is increased 250-fold in phages containing Θrzs. A similarly enriched ORF (140-fold) detected in 54.5% of Θrz-containing phage genomes is annotated as "RNA ligase, DRB0094 family". This enzyme contains a C-terminal adenylyltransferase domain linked to an N-terminal module that resembles aminoacyl-tRNA synthetases[42]. Thus, we propose that it could perform a similar function and may additionally attach the appropriate amino acid to its corresponding tRNA. We assume that the remaining minority of phages likely relies on host-encoded enzymes for these aspects of tRNA maturation.

The non-tRNA-associated Θrz hits (*n* = 526) were categorized based on their closest ORF (Fig. 2c, bottom). The majority (*n* = 493) reside in non-coding regions, most being located more than 200 nt from the nearest annotated ORF. Over 96% of these Θrz hits shared the same directionality (sense) as the closest up- or downstream gene. Only 33 examples were located partially or entirely within an ORF (intragenic). However, the proposed coding regions in these genomes are putative, unverified ORFs, and thus may contain false positives. Considering the anticipated self-cleavage of identified ribozymes in an HDV-like manner, we expect them to be located outside of coding regions. Thus, the frequency of intragenic Θrzs could be used to estimate our false-positive rate more precisely (~2.3%), since phages are known to be very densely coded, yet we still find such low numbers of Θrzs within predicted ORFs. Θrzs in non-coding regions that are not associated with a tRNA likely serve unknown biological functions and may be subjected to future studies. However, tRNA-associated Θrzs constitute the majority of our hit pool and demonstrate the clearest indication of a biological function, prompting us to focus on this subgroup for in vitro validation.

## tRNA-associated Θrzs are active in vitro

The internal transesterification mechanism of drzs relies on an essential cytosine in the J4/2 linker with a perturbed p$K_a$ and a coordinated Mg$^{2+}$ ion, which positions the phosphate backbone for an in-line attack. This acid-base catalyzed reaction has been shown to be most efficient near neutral pH[43]. To validate the HDV-like self-scission of tRNA-associated Θrzs, we selected and investigated four tRNA/Θrz pairs in vitro (Supplementary Fig. 3 and Supplementary Data File 1). We

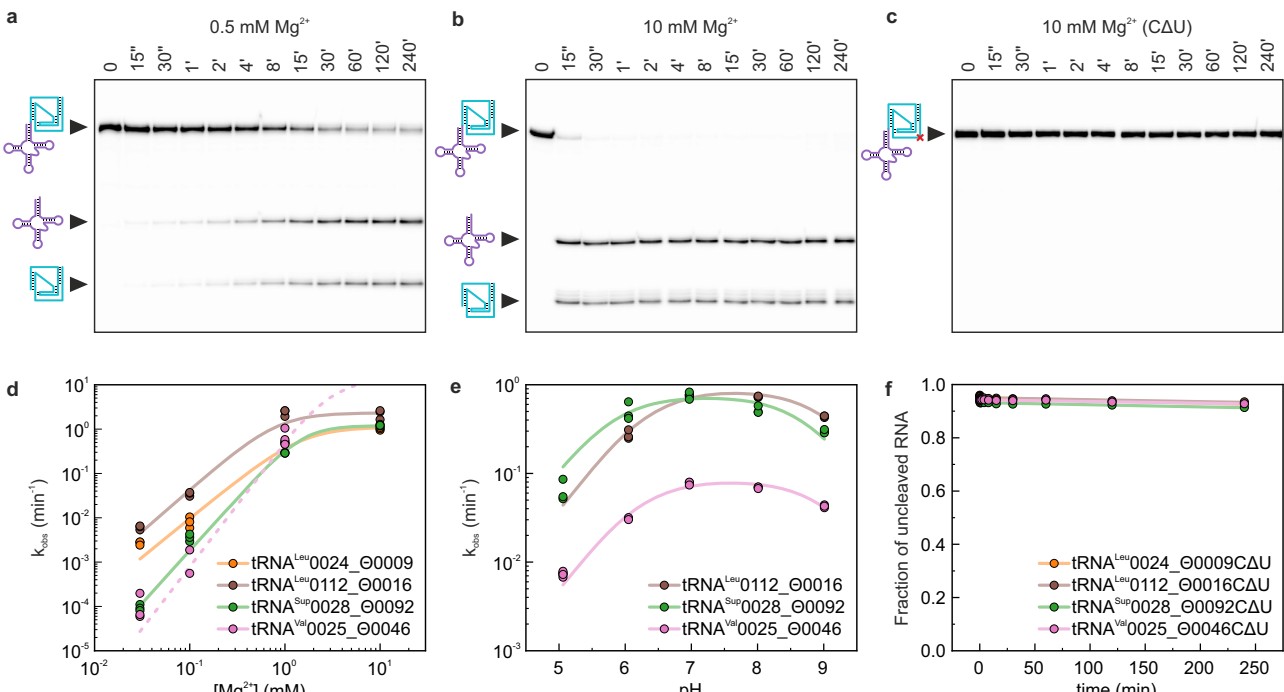

**Fig. 3 | In vitro confirmation of selected tRNA/Θrz pairs. a** Representative PAGE analysis of self-cleavage assays with tRNA$^{Val}$0025_Θ0046 RNA at pH 7.0 and 37 °C induced with 0.5 mM Mg$^{2+}$. Timepoints are indicated above the respective lanes (" = seconds; ' = minutes). Band identities: Precursor RNA = combined cartoon (146 nt); tRNA = purple (90 nt); Θrz = turquoise (56 nt). **b** Representative gel of tRNA$^{Val}$0025_Θ0046 RNA at pH 7.5 and 37 °C induced with 10 mM Mg$^{2+}$. **c** Representative gel of tRNA$^{Val}$0025_Θ0046CΔU RNA at pH 7.5 and 37 °C induced with 10 mM Mg$^{2+}$. **d** Calculated apparent kinetic rate constants ($k_{obs}$) at varying Mg$^{2+}$ concentrations of the four examples tRNA$^{Leu}$0024_Θ0009, tRNA$^{Leu}$0112_Θ0016, tRNA$^{Sup}$0028_Θ0092, and tRNA$^{Val}$0025_Θ0046. The $k_{obs}$-Mg$^{2+}$ dependency for tRNA$^{Val}$0025_Θ0046 at 10 mM Mg$^{2+}$ can only be estimated due to too fast cleavage (dashed line). **e** Calculated $k_{obs}$ at varying pH values of the three examples tRNA$^{Leu}$0112_Θ0016, tRNA$^{Sup}$0028_Θ0092, and tRNA$^{Val}$0025_Θ0046. **f** Relative fraction of precursor RNA of the four inactivated examples (tRNA$^{Leu}$0024_Θ0009CΔU, tRNA$^{Leu}$0112_Θ0016CΔU, tRNA$^{Sup}$0028_Θ0092CΔU, and tRNA$^{Val}$0025_Θ0046CΔU) at 37 °C induced with 10 mM Mg$^{2+}$. Each experiment was performed in triplicates and all individual values were plotted in each graph. Source data are provided as a Source Data file.

chose three pairs based on their high prevalence in our first motif search and the tRNA$^{Val}$0025_Θ0046 pair (for naming see Methods) because of its elongated J4/2 junction, which allowed us to experimentally verify hits obtained by the additionally introduced degree of freedom in the final motif as true positives (Fig. 2b (iv)).

All selected examples exhibited Mg$^{2+}$-dependent self-scission activity in vitro (Fig. 3a,b and Supplementary Fig. 4a). The apparent self-cleavage rate constant ($k_{obs}$) showed a typical HDV-like sigmoidal behavior with increasing Mg$^{2+}$ concentration. Among the constructs, the tRNA/Θrz pair tRNA$^{Val}$0025_Θ0046 exhibited the highest $k_{obs}$ at pH 7.0 (>10 min$^{-1}$ at 10 mM Mg$^{2+}$; Fig. 3b,d), while the other constructs showed significantly slower $k_{obs}$ at the same Mg$^{2+}$ concentration (10 mM; Fig. 3d and Supplementary Table 2).

Titrations of three tRNA/Θrz pairs across a pH range of 5–9 revealed maximal $k_{obs}$ at physiological pH, consistent with previously identified minimal drzs[18] (Fig. 3e and Supplementary Fig. 4b). Two p$K_a$ values were inferred for each construct: p$K_{a1} \approx 9.0$ probably corresponding to a hydrated Mg$^{2+}$ ion and p$K_{a2} \approx 6.0$ related to the catalytic cytosine residue (Supplementary Table 3). Furthermore, we confirmed the inactivation of all tRNA/Θrz pairs upon mutating the catalytic cytosine to uracil (CΔU), in line with known HDV-like behavior[37]. This observation further validates our approach for identifying false-positives in the bioinformatic search (Fig. 3c,f). In summary, these in vitro self-cleavage assays do not show any unexpected behavior but confirm the HDV-like nature of Θrzs.

## Suppressor tRNA-associated Θrzs reveal recoding

To gain insights into the prevalence of Θrzs outside of annotated databases, we conducted an extensive search on raw reads from 469,049 publicly available metagenomic datasets using our improved search motif. This search yielded an additional 104,264 hits (9344 unique Θrz sequences). Despite the short average read length in metagenomic samples (~150–200 bp), 12,515 (12%) of all identified Θrzs were tRNA-associated, resulting in 1698 unique Θrzs adjacent to 5721 unique tRNAs. Consistent with previous findings from bacteriophage databases, the mammalian gastrointestinal tract emerged as the primary environment for Θrzs (Supplementary Fig. 5).

To give a comprehensive overview, tables sorted by Θrz frequency in descending order are provided in Supplementary Data Files 1 and 2. These data include combined Θrz sequences and their adjacent tRNA sequences from annotated and metagenomic samples with additional information such as their taxonomy and predicted bacterial hosts (Supplementary Data File 1, $n$ = 13,257), as well as non-tRNA-associated Θrzs (Supplementary Data File 2, $n$ = 7704). For the sake of completeness, we additionally provide the sequences and genomic coordinates of all minimal drzs discovered in annotated phage databases using the first adaptation (Fig. 2b (i)) of the search motif (Supplementary Data File 4). Remarkably, we found Θrzs associated with predicted tRNAs of all amino acids (Fig. 4a). Among them, 157 unique Θrzs were associated with tRNAs of more than one amino acid isotype (excluding undetermined types where the anticodon-loop could not be assigned unambiguously; Undet), with Θ0013 displaying the greatest diversity (13 different amino acid isotypes).

Alignment and analysis of all unique tRNA-associated Θrz sequences using R2R[44] resulted in a consensus motif (Fig. 4a) featuring several conserved nucleotides not originally defined in the descriptor file. These hits were further categorized based on the associated tRNA type, and over 70% of all hits are associated with either tRNA$^{Met}$, tRNA$^{Sup}$, or tRNA$^{Leu}$ (Fig. 4a). Our interest was piqued especially by the large proportion of Θrzs (20%)

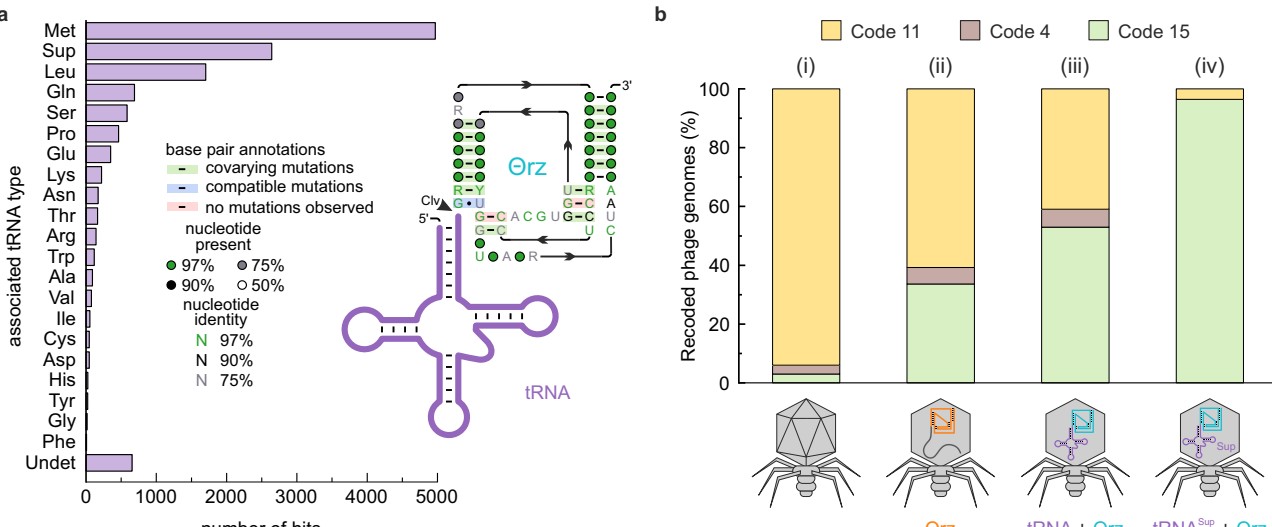

**Fig. 4 | Categorization of tRNA-associated Θrz hits and probability of recoding in phage genomes. a** Total number of Θrz hits ($n = 13,257$) from combined annotated and metagenomic datasets sorted by the type of associated tRNA located ± 5 nt from the Θrz cleavage site. Undet = undetermined tRNA type. A consensus sequence of all unique tRNA-associated Θrz hits ($n = 1,753$) was visualized using R2R[44]. Nucleotide denominations according to IUPAC nomenclature. **b** Fraction of annotated bacteriophage genomes (%) predicted to use code 15 (UAG = Gln; green), code 4 (UGA = Trp; brown), or the standard bacterial genetic code 11 (yellow). (i) The overall expected recoding ratio of phages[23], (ii) phages containing Θrzs (orange), (iii) phages containing tRNA-associated Θrzs (turquoise) adjacent to a tRNA of any type (purple), and (iv) phages containing Θrzs adjacent to tRNA[Sup]. Source data are provided as a Source Data file.

associated with tRNA[Sup], of which 99.7% contain an anticodon for the amber stop-codon (UAG). Intriguingly, 85% of all tRNA[Sup] genes in this subset of phage genomes are associated with a Θrz, i.e., only around 15% tRNA[Sup] genes in these phages lack a downstream ribozyme. The presence of tRNA[Sup] suggests stop-codon reassignments, since tRNA[Sup] are essential elements for the expression of genes containing in-frame stop-codons[23]. These findings may provide answers to a fundamental open question: Why do bacteriophages invest resources in carrying tRNAs instead of utilizing host-provided tRNAs?

Building upon recent findings by ref. 23, who reported stop-codon reassignment in ~2–6% of human and animal gut phages (Fig. 4b (i)), we investigated the genetic codes of the analyzed bacteriophages. Our predictions comprise the predominant recodings code 15 (UAG reassigned to Gln), code 4 (UGA reassigned to Trp), and the standard bacterial code (code 11, no stop-codon reassignments). Genomes with a reassigned stop-codon show gene fragmentation when genes are predicted in standard code. Therefore, we classified a phage genome as recoded if the alternative coding density exceeded a 5–10% increase (depending on the genome size) compared to the coding density in standard code. Remarkably, we found that over one-third (33.7%) of genomes containing Θrz sequences likely utilize code 15, and 5.6% use code 4 (Fig. 4b (ii)). When we narrowed down our analysis to genomes containing tRNA-associated Θrzs, these proportions increased to over half (53.0%) and 6.0%, respectively (Fig. 4b, (iii)). Notably, when analyzing only phage genomes containing tRNA[Sup]-associated Θrzs, a remarkable 96.4% are likely recoded to code 15 (no hits of code 4 were observed; Fig. 4b, (iv)). If these predictions are correct, we would expect these phages to carry tRNA[Sup] with a glutamine isotype. We therefore used a computational isotype prediction model based on bacterial tRNAs[45], which provides an estimate but requires further experimental validation to draw definitive conclusions. This investigation of 180 tRNA[Sup] examples yielded a major fraction (69.4%) with a glutamine isotype, as expected. The remaining tRNA[Sup] show isotypes for Trp (19.4%; rarely reported code 32), Ile (10.0%; unreported recoding), and Tyr (1.1%; code 29). In conclusion, phage genome analysis and tRNA[Sup] isotype predictions both point to code 15 recoding, strengthening the

hypothesis that Θrzs may play a crucial role in the code switch of certain recoded phages.

## Discussion

An iterative improvement of an existing minimal drz motif[18] using bacteriophage genomes resulted in a reliable motif with a low false positive rate and high specificity for tRNA-associated Θrzs (Fig. 2). By restricting the optimization process to unique Θrz sequences we also increased stringency, resulting in the identification of 1753 unique tRNA-associated Θrzs in metagenomic and annotated databases. Although the short length of most raw reads (~150–200 nt) makes the detection of both a Θrz and tRNA (~130-150 nt) on the same read highly unlikely, we still detect a notable proportion (12%) of tRNA-associated examples in metagenomic samples. Thus, the actual number of tRNA-associated Θrzs is likely much higher and probably corresponds approximately to the determined percentage in annotated phage genomes (~58%).

Our biochemical analyses confirmed HDV-like self-scission in vitro for all four selected Θrzs. Moreover, their efficient self-scission rates are comparable to or even faster than those of previously reported metagenomic minimal drz examples[18] (e.g., $k_{obs}$(drz-Mtgn-3) = $1.69 \pm 0.03$ min$^{-1}$ and $k_{obs}$(drz-Mtgn-4) = $0.0022 \pm 0.0001$ min$^{-1}$). These properties make them potential candidates for bioengineering and biomedical applications, such as aptazymes: self-cleaving ribozymes combined with aptamers to control gene expression[7,46–48].

Due to the frequent association of Θrzs with tRNA encoding sequences, we postulate a function in tRNA 3'-trailer processing, which has not been reported to date and is currently limited to *Caudoviricetes* bacteriophages in mammalian gut microbiomes, which appear to infect bacteria of the *Bacteroidota* and *Bacillota* phyla. While the generation of the mature 5'-end of tRNAs is well-understood and usually involves a single ribonucleoprotein enzyme (RNase P) present in all domains of life[49], the 3'-processing of tRNAs is less understood. In *Escherichia coli*, the cleavage of tRNA 3'-trailers involves a complex, multi-step mechanism involving various endo- (RNases E and III) and exonucleases[50] (RNases II, BN, D, PH, PNPase, and T). We postulate that some bacteriophages containing tRNA-associated Θrzs may not need to rely on certain host RNases. By associating a Θrz in *cis* with tRNAs

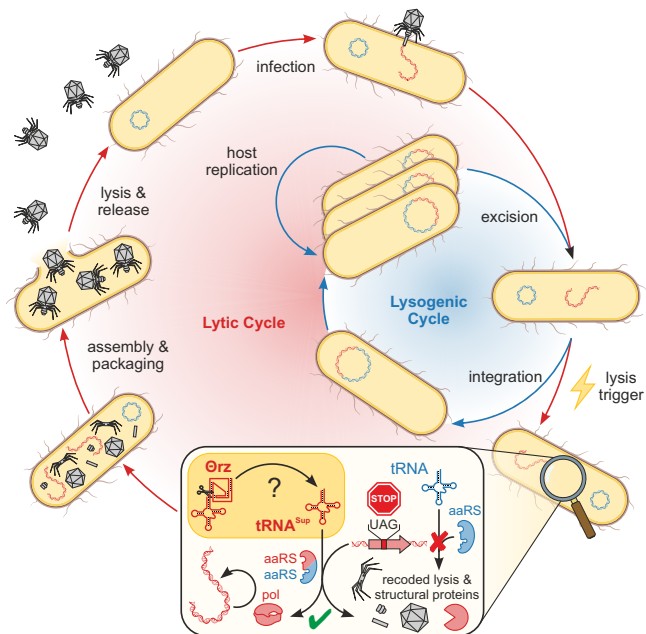

**Fig. 5 | Putative phage infection cycle involving tRNA^Sup-associated Ɵrzs.** A gut bacterium of the *Bacteroidota* or *Bacillota* phylum is infected by a recoded *Caudoviricetes* bacteriophage, which can initiate the lysogenic or lytic cycle. In the lysogenic cycle, the phage genome is integrated into the host genome and host-encoded machinery replicates the host cell including the integrated phage. A not fully understood mechanism triggers the lytic cycle (lightning symbol). Phage-encoded tRNA^Sup, possibly regulated by associated Ɵrzs, enable the translation of recoded lysis and structural proteins (enlarged box). An attempt to produce these proteins using host-encoded tRNAs and aminoacyl synthetases (aaRS) would lead to gene fragmentation (in-frame amber stop-codons). In the last phase of the lytic cycle, phage particles self-assemble, the host cell is lysed, and phage particles are released, leading to a new infection cycle. Polymerase: pol. Colors: Host DNA/proteins: blue; phage DNA/RNA/proteins: red.

instead of encoding a specific nuclease, the phage reduces the genomic space required to produce mature tRNA 3′-ends, despite the need for a ribozyme at each tRNA. In cases where the Ɵrz cleavage site is not directly adjacent to the predicted tRNA 3′-end, we assume that this is likely due to incorrect tRNA prediction by the used software[45]. Otherwise, these could be examples where the phage can utilize its Ɵrz but still needs other RNases to trim the remaining few nucleotides. We hypothesize that this overall reduction of genomic space contributes to phage fitness and opens avenues for regulation.

The regulation of tRNAs is crucial not only for protein biosynthesis, but also for host-manipulation during viral infections: recent evidence has revealed that viral tRNAs can substitute cellular tRNAs and support viral infection[51]. Combined with the requirement of viral tRNAs to sustain translation while the host machinery degrades[52], these factors may explain the prevalence of tRNAs in viral genomes. With respect to tRNA^Sup, which is essential for recoded phages, we show a clear positive correlation between tRNA^Sup-associated Ɵrzs and code 15 phages. The remarkably high abundance and efficiency of Ɵrzs provide additional support and a possible key element in the mechanism recently proposed by ref. 23: tRNA^Sup-associated Ɵrzs might regulate or even initiate the lytic cycle in specific bacteriophages (Fig. 5). Their study highlighted the overrepresentation of stop-codons in phage structural and lysis genes[23], suggesting a pivotal role for stop-codon reassignment in the timing and mechanism of the lysis trigger. Importantly, mistiming can be detrimental, as premature lysis serves as a host defense mechanism. In such a scenario, the host initiates lysis before phage particles are fully assembled within the cell, severely compromising phage efficiency[53–55].

The exact activation and regulation of Ɵrz self-scission as well as the release of the associated tRNA in vivo are still unclear and will be the subject of future studies. An alternative hypothesis not involving the necessity of direct Ɵrz regulation could be that the ribozyme persists in an "always on" state and self-regulates the translation of late phase lytic genes in a concentration-based manner. This would involve highly efficient co-transcriptional tRNA 3′-trailer scission by the Ɵrz, drastically increasing the concentration of tRNA^Sup within the host cell. Once a critical concentration is surpassed, the viral tRNA^Sup overwhelms host-encoded termination factors and leads to the code switch from code 11 to code 15, enabling late-phase viral gene translation and subsequent bacterial host cell lysis. A similar regulation is observed for ribonucleotide reductase, which reduces ribonucleotides to deoxyribonucleotides. This enzyme is activated or deactivated depending on the concentration and, therefore, the preferential binding of ATP or dATP to the allosteric active site, respectively[56,57]. The observed association of Ɵrz sequences with multiple tRNA types further strengthens concentration-based self-regulation. Ɵrzs may not only contribute to a code switch but also optimize the codon usage for phage-encoded genes, further supporting their translation rather than host-encoded genes.

By providing the sequences of all identified Ɵrzs and their associated tRNAs, we offer a valuable resource with thousands of examples for future studies. Furthermore, we report the discovery of tRNA^Sup sequences with a predicted Ile-isotype, representing an unobserved recoding event. Overall, our combined in silico and in vitro results emphasize the importance of ribozymes in biological processes and introduce a subgroup of small tRNA-processing drzs, which may enable extensive tRNA-mediated host manipulations within the mammalian gut microbiome.

## Methods
### Initial BLAST searches
Using the sequence "GGTAGCACACCTATGCGTTCCCGTCGCGCTACT GATTTAGACTAAATAGGT" as a query (drz-Mtgn-1[18]), initial manual homology searches (BLASTn[26]) were conducted against the nucleotide collection databases from NCBI[25]. Several hits with 100% identity were observed, nearby ORFs were screened manually and submitted to further manual homology searches. Flanking regions of ORFs with an *e* value cutoff of 10⁻¹⁰ were investigated by hand initially. Subsequently, the corresponding databases were downloaded and submitted to motif searches using the software RNArobo[27] (see below for detailed description). This procedure was repeated several times, resulting in dozens of hits, among them a first hint of tRNA associations (Supplementary Fig. 1).

### Motif-based database searches and motif improvement
Eukaryotic genomes were chosen from NCBI RefSeq[39] with the following query title on June 10th, 2022:

"Search Eukaryota AND "complete genome"[filter] AND all[filter] NOT anomalous[filter]."

Additionally, human (GCA_000001405.28) and mouse (GCA_000001635.9) genomes were manually added to the genome collection. For Bacteria, the representative genomes from ProGenomes3[38] (https://progenomes.embl.de/download.cgi) were downloaded. The viral databases were obtained on June 10th, 2022, under the links specified in Data availability[23,29–35]. All files were merged for subsequent analysis. Due to some redundancies in the Roux, 2021 database[35] (contained parts of three of our other databases: refs. 31,28,33), only hits that were unique were considered for downstream analysis, i.e., ribozymes that were already detected in other databases were discarded from that source. The RNArobo software[27] v2.1.0 was used to search these databases and define the search motif containing the conserved sequence and structural elements of

minimal drzs. This motif was based on a minimal motif described by ref. 18, and the descriptor file was structured as shown in Supplementary Fig. 6.

The final descriptor file resulted from several iterations of searches on annotated phage genome databases (Supplementary Table 1), where the motif was manually adapted to result in a high percentage of tRNA-associated ribozymes within the total drz hit pool. All databases were searched with RNArobo 2.1.0 and -c --nratio 0.1 parameters 16 times in total: 4 iterations with 4 different motifs (once the active motif and thrice the false-positive motifs). All motif iterations used are depicted in Fig. 2. If a sequence fit into both a false positive as well as true positive motif, it was counted as true positive.

tRNAs were detected using tRNAscan-SE v2.0.9[45] in general mode (-G). Due to computational time constraints, only contigs containing at least one drz hit were screened for tRNAs. Custom code in Python v.3.7.6 was used to combine all outputs and detect Θrzs that are located adjacent to tRNAs. A Θrz is considered tRNA-associated if a tRNA 3′-end was detected within ±5 nucleotides of its cleavage site.

To obtain the predicted isotype tRNAscan-SE 2.0.9 was run in bacterial mode (-B -s), and the output was processed in a script in Python v.3.7.6. All custom code is available under https://github.com/lukasmalfi/theta_ribozymes.

## Metagenomic sequence search

All raw reads from sequence runs available in the MicrobeAtlas project (https://microbeatlas.org) marked as whole genome sequences were used in the subsequent analysis. Raw sequences were downloaded and quality filtered as described under https://microbeatlas.org/index.html?action=help. All 469,049 sample runs matching the criteria were analyzed with both RNArobo and tRNAscan-SE as described above. The sequence read archive (SRA) run IDs of the analyzed samples are stored in Supplementary Data File 3.

The final table containing all tRNA-associated Θrz sequences was constructed using pandas v1.0.3 and NumPy v.1.18.1. Both tables (metagenomic as well as database hits) were combined, and the ribozymes were sorted and named according to their occurrence. For example, tRNA$^{Val}$0025_Θ0046 RNA would be the combination of the 25th-most prevalent tRNA and the 46th-most prevalent tRNA-associated Θrz. Additionally, the source (sample run ID from SRA, or Database source and identifier), their taxonomy (as described later), and predicted hosts (extracted from the metadata of the analyzed databases where available) are provided in Supplementary Data File 1. All unique isolated Θrzs were additionally collected, subsequently named according to their number of occurrences, and uploaded in Supplementary Data File 2. All minimal drz sequences obtained from annotated phage genomes (Supplementary Table 1) with the initial adaptation of the search motif (see Fig. 2b (i)) are available in Supplementary Data File 4.

## Coding density analysis and annotations

Prodigal v2.6.3[58] was used to infer coding sequences based on the three codes 11, 15, and 4 (-g11, -g15, -g4, respectively). The script get_CD.py from ref. 23 (https://github.com/borgesadair1/AC_phage_analysis/releases/tag/v1.0.0) was adapted to calculate coding density for all genomes in the combined database. The coding density of a bacteriophage genome had to be at least 5 or 10% (depending on contig size: <100 kbp: 10%, >100 kbp: 5%) higher with the alternative code than standard code to be considered a recoded phage. These identified ORFs (depending on the predicted code of the phage) were analyzed together with the predicted Θrz sequences to determine the genomic context of non-tRNA-associated Θrzs. Annotations of coding sequences were obtained with geNomad v1.3.3[59] and are provided in Supplementary Data File 5.

## Taxonomy

The taxonomy of all phage genomes was determined using geNomad v1.3.3[59] with the "genomad end-to-end" workflow according to the taxonomy contained in the "International Committee on Taxonomy of Viruses" virus metadata resource number 19. A taxonomy was considered true if at least 75% of the genes agreed in their taxonomic assignments. All contigs containing Θrzs were separately analyzed with the same parameters, and their relative abundances of taxonomic assignments were compared and plotted with OriginPro, Version 2022.

## Word clouds

Samples in the MicrobeAtlas project are annotated with one of four main environments (animal, aquatic, soil, or plant) and keywords extracted from the metadata of the SRA. These environmental assignments and keywords can be found in the file "samples.env.info" obtained from https://microbeatlas.org/index.html?action=download, on March 10th, 2023. Custom code in Python v.3.7.6 was used to extract the environment and all keywords of the respective samples containing at least one tRNA-associated Θrz, excluding the environment of samples annotated with "aquatic, wastewater" since it can be contaminated with human stool samples, thus not accurately representing the aquatic habitat. The list of the obtained keywords was used to create a word cloud with WordCloud v1.5.0[60] with a custom color map and the following parameters:

stopwords = stopwords, prefer_horizontal = 1, min_font_size = 10, max_font_size = 150, relative_scaling = 0.4, width = 1000, collocations = False, height = 400, max_words = 15, random_state = 1, background_color = "white".

Additionally, a "background" expectancy of keywords was generated by repeating the process with 10,000 random metagenomic samples, in addition to analyzing the environment of all samples in the Microbe Atlas project.

## R2R alignments

All unique Θrz sequences were transformed into a Stockholm 1.0 format file using a custom Python 3 script in Jupyterlab v3.5.0. R2R v1.0.6[44] was used with the following parameters: First, the consensus file was created as follows (filenames in square brackets):

--GSC-weighted-consensus [input].sto [consensus_file].cons.sto 3 0.97 0.9 0.75 4 0.97 0.9 0.75 0.5 0.1

Then, the output was generated utilizing a meta file pointing to the consensus file:

--disable-usage-warning [meta_file].r2r_meta [output].pdf

The output.pdf file was edited with CorelDRAW X7 v17.1.0.572.

## DNA template preparation

The plasmid backbone used for in vitro transcription was derived from pJD20, kindly provided by Dr. Anna Marie Pyle. Double-stranded synthetic DNA containing an *EcoR*I restriction site, the T7 promoter sequence (TAATACGACTCACTATA), a transcription start site (GGGAGA) followed by the tRNA/Θrz pair, a *Pst*I restriction site, and a *Hind*III restriction site, was obtained from Azenta Life Sciences. The synthetic DNA was digested with *EcoR*I and *Hind*III and cloned into the plasmid backbone digested with the same restriction enzymes. To obtain the transcription template, the plasmid was either linearized with *Pst*I or the region of interest amplified by PCR using a forward primer (GAATTCTAATACGACTCACTATAGGGA) and individual reverse primers (Supplementary Data File 6).

## RNA in vitro transcription

$^{32}$P body-labeled RNA was prepared by in vitro transcription at 37 °C for 4–5 h in 50-200 μL reaction volumes containing 40 mM Tris-HCl pH 7.5, 40 mM DTT, 2 mM spermidine, 5 mM each ATP, GTP, and UTP, 0.5 mM CTP, 0.01% triton X-100, 10-100 nM template, 10 mM MgCl$_2$, 20–30 μM inhibitor oligo (Microsynth), 0.1–1 μCi/μL [α-$^{32}$P]-CTP

(PerkinElmer and Hartmann Analytic) and an appropriate amount of T7 RNA polymerase (purified in house). The inhibitor oligos were designed individually for each tRNA/Ɵrz pair (CLC Main Workbench v23.0.2) to span the self-cleavage site with a targeted melting temperature of 50 °C to reduce co-transcriptional scission (Supplementary Data File 6).

The reaction was quenched with an equal volume of loading buffer (0.16% bromophenol blue, 10 mM EDTA pH 8.0 in formamide) and loaded onto an 8% denaturing polyacrylamide gel (29:1; Fisher bioreagents). After electrophoresis, the gels were exposed to a phosphorimage screen (FUJI MS 50340272), visualized using a Typhoon FLA 9500 Scanner (control software v1.1) and the bands corresponding to the full-length RNA excised using a clean, sterile scalpel. The RNA was eluted from the crushed gel slice for 4–6 h at 4 °C in five volumes of crush & soak buffer (10 mM MOPS pH 6.0, 1 mM EDTA, and 250 mM NaCl). To precipitate the RNA, 1/10th of the volume of 3 M sodium acetate pH 5.2 and three volumes of ice-cold, absolute Ethanol were added, and the mixture was incubated at −20 °C overnight. The pelleted RNA was washed with 70% ethanol and dissolved in 50 μL ddH$_2$O.

### Self-scission kinetics

Every experiment was carried out in triplicate. The kinetic reaction buffer (140 mM KCl, 10 mM NaCl, and 50 mM Tris-HCl pH 7.5) was pre-warmed at 37 °C prior to the addition of purified $^{32}$P-labeled RNA to a final concentration of 0.5–1 nM after liquid scintillation counting (HIDEX 300SL; MikroWin v4.44). The reaction mixture was equilibrated at 37 °C for 5 min, and self-scission was initiated by the addition of a tenfold concentrated MgCl$_2$ stock solution to the desired concentration. 10 μL aliquots were taken at predetermined time points, quenched with equal volumes of loading buffer, and loaded onto an 8% denaturing PAGE. After electrophoresis, the gel was dried (Whatman Biometra Maxidry D64), exposed onto a phosphorimage screen overnight, and visualized using a Typhoon Scanner. Quantification of the bands was performed using ImageQuant TL v8.2.

In the pH titrations, the kinetic reaction buffer was replaced by two separate three-buffer systems depending on the pH range to guarantee constant ionic strength at all pH values. For pH 4.5–7.5, a buffer containing 25 mM MES (2-(N-morpholino)-ethanesulfonic acid), 25 mM acetic acid, 50 mM Tris (tris(hydroxymethyl)-aminomethane), 10 mM NaCl, and 140 mM KCl was used, whereas for pH 7.5-9.5, a buffer containing 50 mM MES, 25 mM Tris, 25 mM 2-amino-2-methyl-1-propanol, 10 mM NaCl, and 140 mM KCl was used.

The uniformity and correct cleavage site was confirmed by matrix-assisted laser desorption time-of-flight mass spectrometry (MALDI-TOF-MS; Autoflex Speed, Bruker Daltonics) analysis of the co-transcriptionally cleaved Ɵrz extracted from a denaturing PAGE (Supplementary Fig. 7).

### Data fitting

The data fitting was performed using OriginPro, v2022. The band intensities of the cleaved tRNA bands were corrected for the number of cytosine residues and the relative intensities of $f_{tRNA}$ were calculated as follows (Eq. 1):

$$f_{tRNA} = \frac{I_{tRNA}}{I_{tRNA} + I_{substrate}} \tag{1}$$

$I_{tRNA}$ and $I_{substrate}$ correspond to the intensities of the tRNA and substrate (tRNA_Ɵrz RNA), respectively. The obtained values were fit to either an inverted mono- (Eq. 2) or biexponential decay function (Eq. 3):

$$f_{tRNA} = 1 - \left( A * e^{(-k_1*)} + C \right) \tag{2}$$

or

$$f_{tRNA} = 1 - \left( A * e^{(-k_1*t)} + B * e^{(-k_2*t)} + C \right) \tag{3}$$

A, B, and C correspond to the relative fractions of the constructs performing self-scission at the rates $k_1$, $k_2$, or none, respectively. The reported cleavage rates ($k_{obs}$) correspond to $k_1$ from monoexponential and to the faster cleavage rate from biexponential fits.

The cleavage rate–Mg$^{2+}$-relationships were fit to the following Hill-equation (Eq. 4) assuming a single binding event resulting in self-scission with rate $k_1$:

$$k_{obs} = \frac{k_{max}}{1 + \left( \frac{K_d}{[Mg^{2+}]} \right)^n} \tag{4}$$

This allows for the Hill coefficient n.

The cleavage rate–pH-relationships were fit to the following equation (Eq. 5)[18] assuming two titratable groups, namely a hydrated Mg$^{2+}$ ion (p$K_{a1}$) and the catalytic cytosine (p$K_{a2}$):

$$k_{obs} = \frac{k_{max}}{1 + 10^{pH-pK_{a1}} + 10^{pK_{a2}-pH} + 10^{pK_{a2}-pK_{a1}}} \tag{5}$$

### Statistics and reproducibility

No statistical method was used to predetermine the sample size. No data were excluded from the analyses. The experiments were not randomized. The investigators were not blinded to allocation during experiments and outcome assessment.

### Figure preparation

All figures were created in CorelDraw X7 v17.1.0.572. Parts of Figs. 1, 2, 5, and Supplementary Fig. 1 were created with BioRender.com.

### Reporting summary

Further information on research design is available in the Nature Portfolio Reporting Summary linked to this article.

## Data availability

All data and intermediate results required to reproduce the study have been deposited in Zenodo, accessible at https://doi.org/10.5281/zenodo.10299930. The viral databases from open sources were obtained from the following links: https://zenodo.org/record/4776317[29]. http://ftp.ebi.ac.uk/pub/databases/metagenomics/genome_sets/gut_phage_database/[30]. http://www.virusite.org/index.php?nav=download[34]. https://zenodo.org/record/6410225[23]. https://portal.nersc.gov/MGV[31]. https://github.com/RChGO/OVD/[32]. https://datacommons.cyverse.org/browse/iplant/home/shared/iVirus/GOV2.0[33]. https://genome.jgi.doe.gov/portal/IMG_VR/IMG_VR.home.html[35]. The bacterial genomes from ProGenomes3 can be downloaded here: https://progenomes.embl.de/data/repGenomes/progenomes3.contigs.representatives.fasta.bz2[38]. Source data are provided in this paper.

## Code availability

All custom code is uploaded on github and zenodo: https://github.com/lukasmalfi/theta_ribozymes[61].

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

## Acknowledgements

We are grateful to Andrej Luptak, the Sigel and von Mering lab group members for fruitful discussions in the course of this study. We thank the MS core facility of the University of Zurich, Department of Chemistry, led by Laurent Bigler, for performing MALDI-TOF-MS measurements. University of Zurich (C.v.M. KSt. Nr. 74409; R.K.O.S. KSt. Nr. 73521). Swiss National Science Foundation grant 200020_192153 (S.Z.P., R.K.O.S.). Swiss National Science Foundation grant 310030_192569 (L.M., C.v.M.).

## Author contributions

Conceptualization: K.K., L.M., S.Z.P., R.K.O.S. Methodology: K.K., L.M., S.Z.P., S.J., C.v.M., R.K.O.S. Investigation: K.K., L.M. Software: L.M., K.K. Data curation: L.M. Visualization: K.K. Funding acquisition: S.Z.P., R.K.O.S., C.v.M. Project administration: S.Z.P., S.J., R.K.O.S., C.v.M. Supervision: S.Z.P., S.J., C.v.M., R.K.O.S. Writing – original draft: K.K., L.M. Writing – review & editing: K.K., L.M., S.Z.P., S.J., R.K.O.S., C.v.M.

## Competing interests

All authors declare no competing interests.
