## [Peer Review File · Nature Communications]

REVIEWER COMMENTS

Reviewer #1 (Remarks to the Author):

In this manuscript Kienbeck and co-authors report on theta ribozymes, a subfamily of the already described minimal hepatitis delta virus-like ribozymes. These theta ribozymes are often found adjacent to the 3'-end of tRNA genes from bacteriophage genomes assembled from human or mammalian gut metagenomic data. The authors have proven that the tRNA-associated theta ribozymes are able to self-cleave *in vitro* in a similar way as other minimal hepatitis delta virus-like ribozymes. Because these ribozymes are active, they hypothesize that they may be involved in the correct processing of the 3'-trailers of bacteriophage tRNAs. This is a novel biological activity for ribozymes that has not been described before. In addition, the authors find that 20% of theta ribozymes adjacent to tRNAs are found associated with amber stop codon (UAG) suppressor tRNAs. Intrigued by this fact, they analyse genetic code use in bacteriophages and find that half of the genomes containing tRNA-associated theta ribozymes use alternative genetic codes, and that almost all the genomes in which theta ribozymes are associated with suppressor tRNAs are recoded. Since it has been recently shown that stop codon reassignment in bacteriophages may be involved in transition from lysogeny to lysis, the authors posit that tRNA-associated theta ribozymes could also contribute to that switch.

The methodology used in the paper is sound and well explained in the Methods section, except for some minor points that are explained below. The results presented are interesting and relevant, although sometimes what is proven and what is hypothetical is not clearly separated. At the same time, the strength of the findings is overstated at some points (e.g., p. 1, l. 18-19 "theta ribozymes – likely RESPONSIBLE FOR the switch to the expression of recoded lysis and structural phage genes" - that transition has not yet been elucidated and there are most likely other components at play). I also find the name given to theta ribozymes a bit misleading. The first impression upon reading the title is that theta is a novel family of ribozymes rather than a subgroup (those retrieved with the final descriptor) within the family of minimal hdv-like ribozymes. The authors make clear in the text that theta behave like other minimal hdv-like ribozymes, and that their structure is overall very similar. I would suggest changing the name to theta-delta ribozymes or something akin, or making clear in the title that they are hdv-like ribozymes. In addition, it is my notion that what is recoded with the use of suppressor t-RNAs and stop codon reassignment is the phage genome, not the host's, so that should also be corrected in the title.

Some minor points:

p. 2, l. 26: "supports the elucidation of the lytic-lysogenic switch" should be reworded to improve clarity.

p. 5, l. 96 and p. When searching annotated bacteriophage databases with a descriptor optimised to find tRNA-associated theta ribozymes, 58% of theta hits are tRNA-associated. However, when using the same descriptor to search metagenomic datasets only 12% of the theta hits are associated with tRNAs. Is there an explanation for this? Could it be related to the size of the reads, as explained on p. 8, l. 190-193?

p. 7, l. 161: The paper cited shows that suppressor tRNAs are necessary for the expression of genes that contain reassigned stop codons, and that those codons are more prevalent in late structural and lysis genes. That suppressor tRNAs are required for successful phage lysis is not demonstrated in that article.

p. 8, l. 172: Are suppressor tRNAs in the studied bacteriophages all associated with theta ribozymes? Do bacteriophages with tRNA suppressor associated theta ribozymes also contain other suppressor tRNA genes without ribozymes?

Fig 3: It seems in some of the panels that there were different replicates carried out for each combination tRNA-theta and concentration/pH/time, but it is not explained in the Methods section or in the figure legend. That information should be added.

Reviewer #2 (Remarks to the Author):

The manuscript by Kienbeck et al describes discovery of drzs ribozymes associated with tRNA in bacteriophages. These findings are novel and very interesting, with important implications for microbiology and phage biology. The quality of analysis and data presentation is very high. The manuscript is well written and will be of interest to a broad range of readers. I have a few minor comments:

Line 54: Would it be possible to provide taxonomic summary (similar to Extended Data Fig. 1) and sequences/genomic coordinates (similar to Supp. Data File 1) for all drzs, not just the theta-rzs? Refs 28 and 29 point towards phage genome collections, not drzs.

Line 76: "eight recently annotated viral databases" - could the authors name them here or in the supplementary data?

Line 99-100: Could it be because the human and animal gut-derived phage genomes dominated these databases?

Line 104: Is it possible to provide taxonomy of these Caudoviricetes viruses below the class level (family, genus, according to the most recent release of ICTV taxonomy)? What hosts are they associated with (from experimental evidence or by in silico predictions)?

Line 121: "three pairs of higher prevalence" - what phages/bacterial hosts were they associated with?

Reviewer #3 (Remarks to the Author):

Kienbeck & Malfertheimer et al submitted an intriguing study identifying minimal HDV-like ribozymes often associated with tRNA genes in DNA gut phages. Because of their association with tRNAs, they name this self-cleaving ribozyme subclass "theta" ribozymes and raise an interesting hypothesis on their involvement in the recoding of gut phages during the switch between lysogenic and lytic cycle of phages.

The paper is well-written, the figures are neatly arranged, clear and easy to follow with the help of the main text and figure legends.

The methods are described in enough detail to reproduce the data and correspond to the used standards in the field of self-cleaving ribozyme research.

It was a pleasure to read this manuscript and I highly recommend it for publication. However, I do have a few major questions/concerns, I would like to raise before publication.

Main points:

1. One of my main points of concern is whether the claim of theta ribozymes being directly responsible for the switch between lytic and lysogenic cycle and a switch in the expression to recoded lysis and structural phage genes is backed up by enough experimental evidence. Could the authors check for expression patterns in existing phage RNA-seq data that would back up their claims or provide more direct experimental evidence for the switch?

The presented hypothesis how theta ribozymes may be integral to such a switch is sound, but there is no proof of changed expression patterns or ribozyme regulation.

So several claims throughout the paper should be toned down or concrete evidence added.

Line 1: "tRNA-based host recoding in gut phages"

Line 18: "...responsible for the switch to the expression of recoded lysis and structural phage genes"

Line 52: "regulate the code switch in a subset of recoded bacteriophages"

Again, the paper does not provide evidence on a regulation, only on a possible mechanism of tRNA maturation. Based on existing knowledge on small ribozyme activity, cleavage usually occurs immediately upon transcription. The discussion mentions a mechanism taking this "always ON" theory of ribozyme cleavage into account. But again, this should be visible in RNA-seq data, should it not?

2. The authors show that most theta ribozyme occurrences are in intergenic regions, however, they mention in the text (L108-110) and illustrate in Figure 2c that there are a number of ribozymes directly in ORFs. How do they explain this? Are these wrongly annotated ORFs or false positive ribozyme signals? Do these "ORF-ribozymes" show any other properties in comparison to the ones found in intergenic regions? Could the authors please add a brief interpretation/explanation for this observation?

3. In their Abstract (L 22) and elsewhere in the paper (L189), the authors mention over 100 000 examples of theta ribozymes. If I understand it correctly, this number is partly based on (and in my opinion inflated by) the hits found in the raw reads of metagenomic libraries (L142). Such reads are usually highly redundant (thus inflating the hit number, although there are fewer unique examples). Therefore, only unique occurrences should be considered and even they could be somewhat inflated by sequencing errors that do happen and that could create the image of "unique" examples, where there are none, especially in highly abundant RNAs.

I see no necessity for the authors to inflate the number of occurrences like that. Even just taking the hundreds to a few thousands of hits found in annotated genome data bases are convincing to me and create a solid foundation for the other interpretations in their manuscript.

4. Do phage tRNAs encode the necessary CCA-end used for aminoacylation or is it attached by the hosts nucleotidyltransferase machinery? Do the authors find a correlation between an encoded CCA-end and where the ribozyme cleavage site is, e.g. if the CCA is encoded, the ribozyme cleavage site is always 3' of the A in CCA.

The authors state that bacteriophages do not need to rely on host RNases (L207) for tRNA maturation where there are theta ribozymes involved. But then, if the ribozyme cleavage site is up to 5 nt from the discriminator base /end of the conserved tRNA sequence, how do the authors propose the cleavage of these remaining nucleotides occurs, before full tRNA-maturation is ensured? In other words, can one really assume that theta-ribozyme-tRNA combinations where the cleavage site is 5 nt away from the site of CCA-addition are actually functional examples? I would assume for the trimming of the last few nucleotides, they would likely still rely on RNase Z endonucleolytic cleavage, RNase T action or similar. Is there literature evidence that tRNAs with a 5 nt trailer sequence could serve as substrates for CCA-addition? If not, the authors need to soften their claim.

5. L152-154/Figure 4a bottom: I am not clear on what the authors mean by "undetermined tRNA types". Shouldn't it be clear from the anticodon loop? Could they explain what characterizes this group of tRNAs and why they cannot be assigned to an aminoacid?

Minor comments:

When small ribozymes are introduced in the main text, the authors should cite some more recent reviews that include all known classes of these RNAs. The papers cited from around 2006 lack at least 5 classes that have been uncovered since then, so 50% of all known small ribozyme classes.

Possible citations could be:

Egger, M., Bereiter, R., Mair, S., & Micura, R. (2022). Scaling Catalytic Contributions of Small Self-Cleaving Ribozymes. *Angewandte Chemie*, 134(41), e202207590.

Peng, Huan, et al. "Self-cleaving ribozymes: substrate specificity and synthetic biology applications." *RSC chemical biology* 2.5 (2021): 1370-1383.

Weinberg, C. E., Weinberg, Z., & Hammann, C. (2019). Novel ribozymes: discovery, catalytic mechanisms, and the quest to understand biological function. *Nucleic acids research*, 47(18), 9480-9494.

Ren, A., Micura, R., & Patel, D. J. (2017). Structure-based mechanistic insights into catalysis by small self-cleaving ribozymes. *Current opinion in chemical biology*, 41, 71-83.

L122: additionally

L203: ...and usually involves a single ribonucleoprotein enzyme...

There are organisms that catalyze the 5' processing by a protein-only RNase P

L237: allosteric active site

I find the term "active site" more frequently occurs in the literature.

L307: a gut bacterium

L332: is it really June 10th, 2023 or 2022? Later 2022 is mentioned for the accession of other data bases

Editors *Nature Communications*
The Macmillan Campus
4 Crinan Street
London, N1 9XW, UK

Prof Dr. Roland K.O. Sigel
Biological Inorganic Chemistry
roland.sigel@chem.uzh.ch

Prof Dr. Christian von Mering
Bioinformatics
christian.von.mering@mls.uzh.ch

Zurich, October 20th, 2023

***Theta* Ribozymes: A Novel Subgroup of HDV-Like Ribozymes Involved in tRNA-based Recoding of Gut Phages**

by Kasimir Kienbeck[†], Lukas Malfertheiner[†], Susann Zelger-Paulus, Silke Johannsen, Christian von Mering^{*}, Roland K.O. Sigel^{*}

Point-by-Point Response to Reviewers Comments

Note: responses are marked in *green*.

Reviewer #1 (Remarks to the Author)

In this manuscript Kienbeck and co-authors report on *theta* ribozymes, a subfamily of the already described minimal hepatitis delta virus-like ribozymes. These *theta* ribozymes are often found adjacent to the 3'-end of tRNA genes from bacteriophage genomes assembled from human or mammalian gut metagenomic data. The authors have proven that the tRNA-associated *theta* ribozymes are able to self-cleave *in vitro* in a similar way as other minimal hepatitis delta virus-like ribozymes. Because these ribozymes are active, they hypothesize that they may be involved in the correct processing of the 3'-trailers of bacteriophage tRNAs. This is a novel biological activity for ribozymes that has not been described before. In addition, the authors find that 20% of *theta* ribozymes adjacent to tRNAs are found associated with amber stop codon (UAG) suppressor tRNAs. Intrigued by this fact, they analyze genetic code use in bacteriophages and find that half of the genomes containing tRNA-associated *theta* ribozymes use alternative genetic codes, and that almost all the genomes in which *theta* ribozymes are associated with suppressor tRNAs are recoded. Since it has been recently shown that stop codon reassignment in bacteriophages may be involved in transition from lysogeny to lysis, the authors posit that tRNA-associated *theta* ribozymes could also contribute to that switch.

We thank the reviewer for the acknowledgement of our work and the positive assessment thereof.

The methodology used in the paper is sound and well explained in the Methods section, except for some minor points that are explained below. The results presented are interesting and relevant, although sometimes what is proven and what is hypothetical is not clearly separated. At the same time, the strength of the findings is overstated at some points (e.g., p. 1, L18-19 "*theta* ribozymes – likely RESPONSIBLE FOR the switch to the expression of recoded lysis and structural phage genes" - that transition has not yet been elucidated and there are most likely other components at play). I also find the name given to *theta* ribozymes a bit misleading. The first impression upon reading the title is that *theta* is a novel family of ribozymes rather than a subgroup (those retrieved with the final descriptor) within the family of minimal hdv-like ribozymes. The authors make clear in the text that *theta* behave like other minimal hdv-like ribozymes, and that their structure is overall very similar. I would suggest changing the name to *theta*-delta ribozymes or something akin, or making clear in the title that they are hdv-like ribozymes. In addition, it is my notion that what is recoded with the use of suppressor t-RNAs and stop codon reassignment is the phage genome, not the host's, so that should also be corrected in the title.

We thank the reviewer for pointing out the unclear separation of proof and hypothesis, as well as certain overstatements. We followed the reviewer's suggestion and adapted the mentioned sentence (p. 1, L19-20) and sections elsewhere (p. 2, L27-28; p. 3, L52-54; p. 10, L219; p. 11, L258-259; p. 12, L278; p. 13, L285-286) to clarify that theta ribozymes are likely involved in the lytic-lysogenic switch but are not necessarily the cause of it. In the Summary Paragraph, we have clarified that theta ribozymes are a subgroup of minimal HDV-like ribozymes rather than of small ribozymes in general (p. 1, L18-19). The title was changed accordingly. We feel that it is appropriate to keep the name "theta ribozymes" due to their novel biological function (which is unprecedented), inspired by enzymes named after their functions rather than their structural or catalytic properties.

Some minor points:

p. 2, L26: "supports the elucidation of the lytic-lysogenic switch" should be reworded to improve clarity.

We have reworded the sentence to "... introduces a potential new player involved in the code switch of certain recoded gut bacteriophages." (p. 2, L27-28)

p. 5, L96 and p. When searching annotated bacteriophage databases with a descriptor optimized to find tRNA-associated *theta* ribozymes, 58% of *theta* hits are tRNA-associated. However, when using the same descriptor to search metagenomic datasets only 12% of the *theta* hits are associated with tRNAs. Is there an explanation for this? Could it be related to the size of the reads, as explained on p. 8, L190-193?

We thank the reviewer for this comment. Yes, we assume that the size of the reads is the main reason for this observation. The average length of metagenomic reads is approximately 150-200 nts, whereas the average length of combined Θ rz and tRNA is between 130-160 nts or longer. We now included this information in the results part (p. 8, L170-172) and adapted our discussion accordingly (p. 10, L224-229).

p. 7, L161: The paper cited shows that suppressor tRNAs are necessary for the expression of genes that contain reassigned stop codons, and that those codons are more prevalent in late structural and lysis genes. That suppressor tRNAs are required for successful phage lysis is not demonstrated in that article.

We thank the reviewer for this careful observation and fully agree. We have changed the sentence to "... since tRNA^{Sup} are essential elements for the expression of genes containing in-frame stop codons." (p. 9, L195-196)

p. 8, L172: Are suppressor tRNAs in the studied bacteriophages all associated with *theta* ribozymes?

Thank you for this critical question. Due to the computational time constraints, we were unable to search for tRNAs in all genomes of the studied bacteriophages. However, we looked at a subset (all phage genomes containing at least one Θ rz), and around 15% of the tRNA^{Sup} were associated with a Θ rz. This is in line with our expectation that Θ rz together with tRNA^{Sup} are only found in a subset of phages, and not all tRNAs rely on ribozymes for their processing.

Do bacteriophages with tRNA suppressor associated *theta* ribozymes also contain other suppressor tRNA genes without ribozymes?

Excellent question. When analyzing bacteriophage genomes with tRNA^{Sup}-associated Θ rz, a striking 85% of these tRNA^{Sup} are directly adjacent to a Θ rz. We have added this information to the main text (p. 9, L192-194).

Fig 3: It seems in some of the panels that there were different replicates carried out for each combination tRNA-*theta* and concentration/pH/time, but it is not explained in the Methods section or in the figure legend. That information should be added.

Thank you for pointing this out. All experiments were carried out in triplicates and each individual measurement is shown in the graphs of Fig. 3. We now include this information in the figure legend of Fig. 3 (p. 18, L331-332) and the Methods section (p. 26, L492). The exact values of the plotted data points can be seen in Extended Data Table 2 (Mg²⁺ titrations) and Extended Data Table 3 (pH titrations). In addition, we have uploaded all data used in the figures with this revision.

Reviewer #2 (Remarks to the Author)

The manuscript by Kienbeck et al describes discovery of drzs ribozymes associated with tRNA in bacteriophages. These findings are novel and very interesting, with important implications for microbiology and phage biology. The quality of analysis and data presentation is very high. The manuscript is well written and will be of interest to a broad range of readers.

We are very grateful to the reviewer for the positive remarks and the careful review. We appreciate the thoughtful comments.

I have a few minor comments:

p. 3, L54: Would it be possible to provide taxonomic summary (similar to Extended Data Fig. 1) and sequences/genomic coordinates (similar to Supp. Data File 1) for all drzs, not just the *theta*-rzs?

*Thank you for these suggestions. We have added a taxonomic summary of all drzs in Extended Data Fig. 2b. However, we would like to explicitly state that the confidence for the prediction of some of the non-*theta* drzs is low, i.e., there might be a high false-positive rate (as shown in Fig. 2b (i)). Regarding the sequences and genomic coordinates of all drzs, we have added an additional file (Supplementary Data File 4) with this information for all ribozymes with the drz motif shown in Fig. 2b (i) that were found in annotated phage genomes. We did not add additional information for metagenomic drzs due to the high number of false positives.*

Refs 28 and 29 point towards phage genome collections, not drzs.

(Please note that these references have shifted and are now 30 and 31, respectively.) This is correct. If you are referring to p. 3, L66-67, we are explaining our very first attempt at applying a motif-based search using a minimal drz motif rather than a sequence-based BLAST search using the sequence of drz-Mtgn-1. In this initial search, we got around 60 hits in the databases of Refs 30 and 31, which caused us to expand our search to more databases, including metagenomic sequencing data. We have adapted the text in this section accordingly (p. 3, L63-67).

p. 4, L76: "eight recently annotated viral databases" - could the authors name them here or in the supplementary data?

These databases are listed and referenced in Extended Data Table 1, as indicated by the parentheses. We have additionally added the references at the mentioned position (p. 4, L79).

p. 5, L99-100: Could it be because the human and animal gut-derived phage genomes dominated these databases?

Thank you for this critical question. Indeed, half of the databases (4 out of 8) originate from human or animal gut, the other half are from diverse environments (see Extended Data Table 1). However, the mammalian gut-derived databases account for only 24.4% (34.8 billion base pairs; Gbp) of the total base pairs of all 8 databases (142.8 Gbp). Nevertheless, these databases yield the vast majority (679 hits, 91.6%) of the total tRNA-associated Θ rz hits within these 8 databases (741 hits). To verify that

our hits are indeed enriched in mammalian gut-derived phage genomes, we had included two non-viral databases as a control (bacterial and eukaryotic genomes), totaling 318.4 Gbp, i.e., more than twice the size of all 8 databases combined, which did not yield a single tRNA-associated Θ rz hit. This result gives us high confidence that the enrichment in gut phages is unbiased. A similar trend is observed in the analyses of metagenomic samples, depicted in Extended Data Fig. 5. While it is true that the majority of the analyzed samples are animal-derived, we also have a sizeable fraction of samples from other environments. Despite this, the samples in which Θ rz are detected are almost exclusively from animal samples.

We have adapted the text to reflect this more clearly (p. 5, L105-109).

p. 5, L104: Is it possible to provide taxonomy of these *Caudoviricetes* viruses below the class level (family, genus, according to the most recent release of ICTV taxonomy)? What hosts are they associated with (from experimental evidence or by *in silico* predictions)?

We used a recent version of the ICTV taxonomy for our analysis (VMR number 19), and the categorization we obtained was the most detailed (some examples were classified further to Crassvirales) we could derive from the program we used (geNomad). We have included this information in the methods (p. 24, L431-433) as well as added the taxonomy of the respective phages to the Supplementary Data File 1.

*Additionally, some of the databases already predicted their hosts *in silico*, in these cases (approximately 35%) we extracted the relevant information and added it to Supplementary Data File 1. From these phages, it appears that the majority infect bacteria of the Bacteroidota phylum, while the rest appears to have Bacillota as preferred hosts. We also added this information into the results (p. 5, L112-114)*

p. 6, L121: "three pairs of higher prevalence" - what phages/bacterial hosts were they associated with?

*The three ribozymes were among the first 60 hits mentioned in p. 3, L66-67 and turned out to be extremely abundant in the overall search (9th-, 16th-, and 93rd-most prevalent tRNA-associated Θ rz found, respectively). They were all associated with bacteriophages of the Caudoviricetes class (with some more detailed taxonomic assignments pointing to the Crassvirales order). Some phages containing Θ 0009 are predicted to infect *Paraprevotella*, a subset of phages with Θ 0016 and Θ 0092 infect *Prevotella*, whereas those containing Θ 0046 have *Parabacteroides* as predicted hosts. We have now added both the taxonomy of the phages as well as their putative hosts to Supplementary Data File 1, where interested readers can look up all predictions of the pairs (using the respective Θ rz and tRNA numbers) in detail.*

Reviewer #3 (Remarks to the Author)

Kienbeck & Malfertheiner *et al.* submitted an intriguing study identifying minimal HDV-like ribozymes often associated with tRNA genes in DNA gut phages. Because of their association with tRNAs, they name this self-cleaving ribozyme subclass "*theta*" ribozymes and raise an interesting hypothesis on their involvement in the recoding of gut phages during the switch between lysogenic and lytic cycle of phages.

The paper is well-written, the figures are neatly arranged, clear and easy to follow with the help of the main text and figure legends.

The methods are described in enough detail to reproduce the data and correspond to the used standards in the field of self-cleaving ribozyme research.

It was a pleasure to read this manuscript and I highly recommend it for publication.

We thank the reviewer very much for these encouraging remarks, and for the detailed review that follows.

However, I do have a few major questions/concerns, I would like to raise before publication.

Main points:

1. One of my main points of concern is whether the claim of *theta* ribozymes being directly responsible for the switch between lytic and lysogenic cycle and a switch in the expression to recoded lysis and structural phage genes is backed up by enough experimental evidence. Could the authors check for expression patterns in existing phage RNA-seq data that would back up their claims or provide more direct experimental evidence for the switch? The presented hypothesis how *theta* ribozymes may be integral to such a switch is sound, but there is no proof of changed expression patterns or ribozyme regulation.

We appreciate this excellent point. We have searched raw RNAseq reads totaling over 30 Terabytes of data in a similar fashion as the metagenomic analyses. In total, we observed ~650 unique Θ z sequences (~22,000 hits in total), of which only 15 hits are directly associated with a tRNA. Additionally, we only find very rare cases (96) of a Θ z directly at the 5'-end of the read, which would indicate cleavage. We assume that the vast majority of these ~22,000 Θ z hits are part of the ~40% non-tRNA associated examples that may not cleave, are regulated by unknown factors, or may cleave at a later timepoint.

At first, we were surprised by this observation, but literature research showed that conventional linker ligation in RNAseq requires a 5'-phosphate and/or a 3'-hydroxyl group. Notably, RNA ends resulting from small self-cleaving ribozymes have a 5'-hydroxyl group and a 2',3'-cyclic phosphate, which are both not recognized as ligase substrates. A study from 2021 has proposed a method capable of sequencing these types of transcripts (DOI: 10.1080/15476286.2021.1999105), but we were unable to find suitable datasets generated by this method. A second point to consider is that generally, tRNAs are hard to find in RNAseq data since they are highly modified and the resulting interference with Watson-Crick base-pairing of the reverse transcription enzyme hinders correct and/or complete reverse transcription. A new method claims to tackle this issue (DOI: 10.1038/s41587-023-01743-6), however, we also could not find any suitable data generated by this novel approach yet. To summarize, we are very grateful for this proposed idea, however, we think that the results we obtained are inconclusive and therefore decided not to include this information in our manuscript. Instead, we have decided to tone down our claims that Θ z might be directly responsible for the lytic-lysogenic switch, now discussing more towards a crucial involvement of Θ z in this switch, where multiple still unknown factors play a role (p. 1, L19-20; p. 2, L27-28; p. 3, L52-54; p. 10, L219; p. 11, L258-259; p. 12, L278; p. 13, L285-286).

So several claims throughout the paper should be toned down or concrete evidence added.

p. 1, L1: "tRNA-based host recoding in gut phages"

The title has been changed to "... involved in tRNA-based recoding of gut phages"

p. 1, L18: "...responsible for the switch to the expression of recoded lysis and structural phage genes"

This line has been changed to "... potentially involved in the switch leading to the expression of recoded lysis and structural phage genes."

p. 3, L52: "regulate the code switch in a subset of recoded bacteriophages"

This line has been changed to "... may support expression of late-phase lysis and structural genes containing recoded stop-codons in a subset of recoded bacteriophages, ..."

Again, the paper does not provide evidence on a regulation, only on a possible mechanism of tRNA maturation. Based on existing knowledge on small ribozyme activity, cleavage usually occurs immediately upon transcription. The discussion mentions a mechanism taking this "always ON" theory of ribozyme cleavage into account. But again, this should be visible in RNA-seq data, should it not?

Thank you for this excellent idea, however, we believe we will run into similar problems as mentioned above in our response under point 1 from this reviewer.

2. The authors show that most *theta* ribozyme occurrences are in intergenic regions, however, they mention in the text (p. 5, L108-110) and illustrate in Figure 2c that there are a number of ribozymes directly in ORFs. How do they explain this? Are these wrongly annotated ORFs or false positive ribozyme signals? Do these "ORF-ribozymes" show any other properties in comparison to the ones

found in intergenic regions? Could the authors please add a brief interpretation/explanation for this observation?

Thank you for this crucial observation, we assume that the explanation is indeed a mix of your proposed reasoning. We have tried to limit false positives in our motif search to a minimum, yet we still find a low number of false positive hits with the final search motif (Fig. 2b (iv)). Concerning the ORFs, most of these are putative ORFs predicted from metagenomic genome assemblies and not experimentally confirmed. Therefore, we assume some false-positive predictions on this side as well. Since all Θ zrs were discovered using the same motif, they have similar properties. When looking at annotated phage genomes, we only find 33 ribozymes within predicted ORFs (out of a total of ~1,300 Θ zrs found in annotated genomes). This number could be used to estimate our false positive rate more precisely (~2.3%) since phages are known to be very densely coded, yet we still find such low numbers within proposed coding regions. We have changed the relevant section accordingly (p. 6, L131-138).

3. In their Abstract (p. 1, L22) and elsewhere in the paper (p. 8, L189), the authors mention over 100,000 examples of *theta* ribozymes. If I understand it correctly, this number is partly based on (and in my opinion inflated by) the hits found in the raw reads of metagenomic libraries (p. 6, L142). Such reads are usually highly redundant (thus inflating the hit number, although there are fewer unique examples). Therefore, only unique occurrences should be considered and even they could be somewhat inflated by sequencing errors that do happen and that could create the image of "unique" examples, where there are none, especially in highly abundant RNAs.

I see no necessity for the authors to inflate the number of occurrences like that. Even just taking the hundreds to a few thousands of hits found in annotated genome data bases are convincing to me and create a solid foundation for the other interpretations in their manuscript.

We thank the reviewer for pointing this out and we agree. We do believe that providing all 100,000 examples (including duplications) can be valuable to the research community (e.g., to see in which samples they were found, with what frequency, etc.), however, it might indeed be inappropriate to use that number in the abstract and elsewhere. Hence, we followed the reviewer's suggestion and have changed the respective mentions to reflect a conservative amount of 1,753 unique tRNA-associated Θ rz occurrences (p. 2, L25; p. 10, L224).

4. Do phage tRNAs encode the necessary CCA-end used for aminoacylation or is it attached by the hosts nucleotidyltransferase machinery? Do the authors find a correlation between an encoded CCA-end and where the ribozyme cleavage site is, e.g., if the CCA is encoded, the ribozyme cleavage site is always 3' of the A in CCA.

*We thank you for this critical question. We have reanalyzed our tRNA sequences associated with Θ zrs and found that only 5% of predicted tRNAs encode a CCA-end. It should be noted that this could correlate with the hosts of the phages, since CCA-end encoding is organism-dependent, e.g., they are generally encoded in gram-negative bacteria such as *E. coli*, but only sporadically encoded in gram-positive bacteria such as *B. subtilis*. When analyzing the position of the cleavage site within this subset of CCA-encoding tRNAs, we find nothing out of the ordinary: 75% have the cleavage site directly 3' of the A in the CCA-tail, in accordance with our general findings (77%). Inspired by this comment, we checked the phage genomes containing Θ zrs for the presence of CCA-adding enzymes. Remarkably, 20% of these phage genomes contain tRNA adenylyltransferase genes, the enzyme responsible for adding a CCA-tail to tRNAs. When analyzing our complete collection of searched bacteriophage genomes, only 0.08% carry this enzyme, i.e., this enzyme ORF is increased 250-fold in phages containing Θ zrs. A similarly enriched protein (140-fold) detected in 54.5% of Θ rz-containing phage genomes is annotated as "RNA ligase, DRB0094 family", which is known to contain a C-terminal adenylyltransferase domain linked to an N-terminal module that resembles aminoacyl-tRNA synthetases (DOI: 10.1074/jbc.M407657200). Thus, we propose it could also perform a similar function and potentially attach the appropriate amino acid to its corresponding tRNA. We have added this information to the main text (p. 5, L114-p. 6, L126).*

The authors state that bacteriophages do not need to rely on host RNases (p. 9, L207) for tRNA maturation where there are *theta* ribozymes involved. But then, if the ribozyme cleavage site is up to 5 nt from the discriminator base /end of the conserved tRNA sequence, how do the authors propose the cleavage of these remaining nucleotides occurs, before full tRNA-maturation is ensured? In other words, can one really assume that *theta*-ribozyme-tRNA combinations where the cleavage site is 5 nt away from the site of CCA-addition are actually functional examples? I would assume for the trimming of the last few nucleotides, they would likely still rely on RNase Z endonucleolytic cleavage, RNase T action or similar. Is there literature evidence that tRNAs with a 5 nt trailer sequence could serve as substrates for CCA-addition? If not, the authors need to soften their claim.

We agree with the reviewer that this claim is stated perhaps too strongly. However, it should be noted that the majority of tRNA-associated Orzs (77%) are located precisely at the predicted 3'-end of their respective tRNA. We assume that the 1-5 nucleotide distance is a bioinformatic artifact, i.e., the tRNA 3'-end is not always predicted with 100% accuracy by the used software (tRNAscan-SE). We assume that certain enzymes, including RNases, may well be essential for bacteriophages containing Orzs. However, they might be able to function without certain specific endonucleases and an adenylyltransferase, while still having to rely on other tRNA 3'-processing enzymes. This might be true specifically in cases where the CCA-tail is encoded with a Orz at the correct position. We have softened the claim and elaborated more on this matter in the mentioned section (p. 11, L243-251).

5. p. 7, L152-154/Figure 4a bottom: I am not clear on what the authors mean by "undetermined tRNA types". Shouldn't it be clear from the anticodon loop? Could they explain what characterizes this group of tRNAs and why they cannot be assigned to an amino acid?

This is an output of the software we used to predict tRNAs (tRNAscan-SE). We did not find a clear explanation for this in their documentation, but we assume that these structures are low-confidence tRNAs where the anticodon-loop could not be unambiguously assigned. This could especially be true for metagenomic reads for which sequencing quality may be poor. We have added this information in the mentioned section of the text (p. 8, L184-185).

Minor comments:

When small ribozymes are introduced in the main text, the authors should cite some more recent reviews that include all known classes of these RNAs. The papers cited from around 2006 lack at least 5 classes that have been uncovered since then, so 50% of all known small ribozyme classes. Possible citations could be:

Egger, M., Bereiter, R., Mair, S., & Micura, R. (2022). Scaling Catalytic Contributions of Small Self-Cleaving Ribozymes. *Angewandte Chemie*, 134(41), e202207590.

Peng, Huan, et al. "Self-cleaving ribozymes: substrate specificity and synthetic biology applications." *RSC chemical biology* 2.5 (2021): 1370-1383.

Weinberg, C. E., Weinberg, Z., & Hamman, C. (2019). Novel ribozymes: discovery, catalytic mechanisms, and the quest to understand biological function. *Nucleic acids research*, 47(18), 9480-9494.

Ren, A., Micura, R., & Patel, D. J. (2017). Structure-based mechanistic insights into catalysis by small self-cleaving ribozymes. *Current opinion in chemical biology*, 41, 71-83.

Thank you for this remark, we have replaced three outdated references with the proposed references (p. 2, L32-34).

L122: additionally

L203: ...and usually involves a single ribonucleoprotein enzyme...

There are organisms that catalyze the 5' processing by a protein-only RNase P

L237: allosteric active site

I find the term "active site" more frequently occurs in the literature.

L307: a gut bacterium

L332: is it really June 10th, 2023 or 2022? Later 2022 is mentioned for the accession of other databases?

We thank the reviewer for the very detailed revision of our manuscript. We have corrected the mentioned mistakes.

REVIEWERS' COMMENTS

Reviewer #2 (Remarks to the Author):

I'm satisfied with the response to my comments and the improvements made to the manuscript. I have no further comments.

Reviewer #3 (Remarks to the Author):

The authors have addressed all my comments and concerns.